# Computational Development of Inhibitors of Plasmid-Borne Bacterial Dihydrofolate Reductase

**DOI:** 10.3390/antibiotics11060779

**Published:** 2022-06-07

**Authors:** Pedro J. Silva

**Affiliations:** 1FP-I3ID, FP-BHS, Faculdade de Ciências da Saúde, Universidade Fernando Pessoa, 4200-150 Porto, Portugal; pedros@ufp.edu.pt; 2UCIBIO@REQUIMTE, BioSIM, Departamento de Biomedicina, Faculdade de Medicina, Universidade do Porto, 4200-319 Porto, Portugal

**Keywords:** computer-aided molecular design, molecular dynamics, density-functional theory, molecular docking, drug development

## Abstract

Resistance to trimethoprim and other antibiotics targeting dihydrofolate reductase may arise in bacteria harboring an atypical, plasmid-encoded, homotetrameric dihydrofolate reductase, called R67 DHFR. Although developing inhibitors to this enzyme may be expected to be promising drugs to fight trimethoprim-resistant strains, there is a paucity of reports describing the development of such molecules. In this manuscript, we describe the design of promising lead compounds to target R67 DHFR. Density-functional calculations were first used to identify the modifications of the pterin core that yielded derivatives likely to bind the enzyme and not susceptible to being acted upon by it. These unreactive molecules were then docked to the active site, and the stability of the docking poses of the best candidates was analyzed through triplicate molecular dynamics simulations, and compared to the binding stability of the enzyme–substrate complex. Molecule **32** ([6-(methoxymethyl)-4-oxo-3,7-dihydro-4*H*-pyrano[2,3-d]pyrimidin-2-yl]methyl-guanidinium) was shown by this methodology to afford extremely stable binding towards R67 DHFR and to prevent simultaneous binding to the substrate. Additional docking and molecular dynamics simulations further showed that this candidate also binds strongly to the canonical prokaryotic dihydrofolate reductase and to human DHFR, and is therefore likely to be useful to the development of chemotherapeutic agents and of dual-acting antibiotics that target the two types of bacterial dihydrofolate reductase.

## 1. Introduction

Inhibition of dihydrofolate reductase (DHFR) is lethal to most cells, as it prevents the synthesis of tetrahydrofolate, which plays a crucial role as a methyl donor in the synthesis of thymidylate from uridylate. Several competitive inhibitors of dihydrofolate reductase are therefore used as chemotherapeutic agents (e.g., methotrexate) or as antibiotics (e.g., trimethoprim, which binds much more strongly to the bacterial DHFR than to human DHFR [1] and is therefore non-toxic for human cells). Several bacterial taxa acquire resistance to trimethoprim through the acquisition of plasmid-encoded dihydrofolate reductases [2,3,4], at least one of which (R67 DHFR, or Type II DHFR) bears no structural resemblance to classical dihydrofolate reductases [5]. R67 DHFR is a soluble homo-tetramer which contains a single symmetrical pore traversing the length of the molecule, lined by amino acids from all four monomers and where folate and NADPH bind. The high symmetry of the active-site-containing pore implies that evolution of catalytic ability in R67 DHFR should face large constraints, as each mutation in the DHFR gene leads to either no changes around the active site (if it occurs away from the pore-facing surface) or to four simultaneous, symmetrical, changes which may have quite contradictory effects on the ability of binding each of the substrates [6]. Individual mutations in the crucial V66-Q67-I68-Y69 substrate-binding region are therefore most often deleterious [7,8,9]. Surprisingly, combinatorial exploration of the pore surface identified several instances where multiple simultaneous mutations of the substrate-binding region afforded native-like activity, whereas changing one of the amino acids in these mutants to the wild-type residue resulted in total loss of activity [10]. Besides confirming the exquisite sensitivity of the active site to individual changes, these data showed that catalysis by R67 DHFR does not require the active intervention of specific amino acid sidechains (e.g., as proton donors/acceptors), but is rather controlled by the spatial arrangement of folate vis à vis NADPH allowed by the pore geometry, and by the general electrostatic environment of the cavity itself [10]. The non-intervention of specific amino acids as catalytic aides is likely to be responsible for its relatively modest catalytic activity, which is 200-fold lower [11] than that of the highly optimized chromosomally encoded enzyme [12].

Due to the remarkable differences in selectivity of the different classes of DHFR, the development of drugs targeting the trimethoprim and methotrexate-resistant, plasmid-borne, bacterial R67 dihydrofolate reductases may enable potent antibiotic activity accompanied by low toxicity to human cells. In spite of this, limited effort has been expended so far in this endeavor which has resulted in the discovery of a single class of symmetrical competitive inhibitors based on 1*H*-benzimidazole-5-carboxylic acid [13,14] with low toxicity towards mammalian cells, but limited potential for use as drugs due to relatively low affinity towards the target DHFR (K_i_ = 2–4 µM, instead of the nanomolar range usually required for a successful drug). The present paper describes a computational search for dihydrofolate analogues, which may competitively bind to DHFR. Molecular dynamics simulations of the best candidates show that their interaction with the enzyme is better than that of the natural substrate, strongly suggesting that they may be suitable for further development as lead compounds for novel DHFR-targeting drugs. Moreover, the best candidate is shown to also bind strongly to the chromosomally encoded “regular” DHFR, as well as to human DHFR, enabling it to be further improved into broader-spectrum antibacterial and chemotherapeutic agents.

## 2. Results and Discussion

### 2.1. Computational Development of Unreactive Analogues of Folate

Direct hydride transfer from NADPH to non-protonated folate has been shown to be prohibitively expensive in the canonical dihydrofolate reductase from *E. coli* [15], therefore requiring substrate reduction to be preceded by protonation of the substrate N^5^-atom. The same energetic constraints are present in the plasmid-borne dihydrofolate reductase, where substrate turnover has moreover been shown to require the substrate N^5^-atom to be protonated by solvent, due to the absence of a proton donor in the active site [16]. Development of a substrate analogue into an effective competitive inhibitor therefore requires that, besides having a high affinity for the active site of the enzyme, the selected substrate analogue should be not only resistant to protonation in its N^5^-position but also unreactive towards NADPH in its N^5^-deprotonated form, to prevent it from reacting with NADPH in any of its potential protonation states. The required resistance to N^5^-protonation in turn implies that, in order to act as an inhibitor, the candidate must be able to bind to the plasmid-borne dihydrofolate reductase in the deprotonated state. The successful design of putative inhibitors therefore required us to ascertain the proton affinity and redox potential of each potential analogue.

Candidate analogues of the dihydropterin core were designed through sequential substitution of the heteroatoms in the pteridine system, as well as replacement of the amine substituent by neutral groups (aiming at obtaining less reactive analogues) or positively charged groups (to enable better binding through interactions with the phosphate groups in the NADPH) (Figure 1).

Quantum chemical computations were then performed as described in the Methods Section. Most substitutions tested afforded less basic molecules than the original folate (Table 1). Interestingly, the effect of most substitutions proved to be additive: for example, the extra basicity of molecule **4** (which has been mutated from dihydropterin in two positions) is almost exactly predicted by the individual effects of each of the individual mutations (**2** and **3**). The only exceptions to this pattern were observed in the molecules where N^8^ had been replaced by O simultaneously with the replacement of N^5^ by CH. In contrast to what was observed for the protonation affinities, two thirds of the substitution patterns tested were found to facilitate the reduction of the pteridine analogues, and in a third of the cases even decreasing the reduction energy to values below 22 kcal.mol^−1^, which is the theoretical limit below which their reduction by NADPH can be performed at reasonable rates (1 h^−1^) at room temperature.

The intersection of the sets of molecules predicted to have much lower basicity than dihydropterin and to strongly resist reduction by NADPH afforded seven hits with suitable chemical inertness: **11**, **15**, **20**–**23**, **31** and **32**. Docking of these molecules into the active site of NADPH-bound R67 dihydrofolate reductase suggested (Table 2) that the positively charged molecules **31** and **32** were the most likely to afford better binding than the dihydropterin core of dihydrofolate (**1**). Molecular dynamics simulations of those complexes were therefore performed and compared with similar simulations of the predicted complexes bearing either dihydropterin or the full folate molecule.

### 2.2. Binding Stability of the Unmodified Dihydropterin Core (Molecule ***1***)

The deprotonated form of the dihydropterin core (molecule **1**) was surprisingly shown by our MD simulations to have remarkably low affinity towards the R67 DHFR active site: it completely separated from the protein in less than 30 ns in two of the triplicate simulations, and disengaged before 50 ns had elapsed in the last simulation (Figure 2). Although N^5^-protonation of this molecule moderately increased its affinity to the active site and enabled it to remain bound to it and in close proximity to NADPH in two out of three simulations (Figure 3), these results clearly show that the dihydropterin core is, by itself, not very prone to interact with R67 dihydrofolate reductase, especially in the deprotonated state. The productive binding of folate with R67 DHFR needed for catalysis to occur must therefore be primarily due to other factors, such as the interactions of its *p*-aminobenzoylglutamate tail with specific features of the protein surface, as already hinted at by previous experimental studies [17].

### 2.3. Binding Dynamics of Folate in the R67 DHFR Channel

To ascertain the role of the benzoylglutamate tail in folate binding, we performed additional simulations with the complete ternary R67 DHFR:folate:NADP^+^ complex in both the N^5^-protonated (Figure 4) and the N^5^-deprotonated (Figure 5) states. As expected from the experimental observations, folate remained bound to the protein, but the simulations revealed considerable variation in binding mode throughout (and between) simulations, with remarkably little reliance on specific interactions such as those between the negatively charged benzoylglutamate tail and the two positively charged Lys32 residues present at the entrance of the pore: indeed, we found that the distances between those groups varied much more widely than observed in previous (shorter) molecular dynamics/empirical valence bond studies of the complex with protonated folate [18]. Our simulations are therefore in better agreement than previous studies with the lack of well-defined binding orientation of the glutamate tail observed in NOE [19] and crystallographic studies [5,20].

Interestingly, whereas the N^5^-protonated dihydrofolate is known to be the catalytically active substrate, the binding profiles are actually much more stable for the deprotonated dihydrofolate: in this form, the reactive dihydropterin C^6^ and nicotinamide C^4^ atoms consistently lie around 4 Å from each other (Figure 5), whereas in the simulations of the N^5^-protonated dihydrofolate the extra positive charge on the dihydropterin is attracted towards the negatively charged phosphate present in the 2′-position of the NADP^+^ ribose ring, quickly leading to larger distances between the dihydropterin C^6^ and nicotinamide C^4^ atoms, and in one of the simulations of the N^5^-protonated dihydrofolate (Figure 4A,D) the substrate was actually seen to bend over itself. Regardless of whether this bending represents a frequent event or is an artifact caused by electrostatic repulsion between the protonated dihydropterin core and the oxidized NADP^+^ present in our simulations, the other five simulations establish the better capacity of the deprotonated substrate to achieve a productive conformation in the active site. It therefore appears that, in spite of the requirement of protonation of the ligand by the solvent, the structure of the active site forces this protonation to occur only after substrate binding, in an active site that is devoid of proton donors. This factor may contribute to the low k_cat_/K_M_ of this enzyme (≈1000 times lower than that of its chromosomally encoded counterpart [6]): the enzyme-substrate complex is only stable when the substrate is in the deprotonated (unreactive) form, but since catalytic competence can only be achieved upon protonation and the active site is bereft of proton-donating sidechains, substrate conversion will only occur during the small portion of ligand dwelling time in the active site when a rare hydronium from bulk solution coincidentally wanders into it.

### 2.4. Binding Dynamics of Candidates ***31*** and ***32*** in the R67 DHFR Channel

Molecules **31** ([6-(methoxymethyl)-4-oxo-3,7-dihydro-4*H*-pyrano[2,3-d]pyrimidin-2-yl]ethylamine) and **32** ([6-(methoxymethyl)-4-oxo-3,7-dihydro-4*H*-pyrano[2,3-d]pyrimidin-2-yl]methyl-guanidinium) were designed with a positively charged substituent on C_2_ to enable them to dock more strongly into the active site through electrostatic interactions with the anionic diphosphate present in NADPH. This strategy proved fruitful, since both candidates remained tightly bound throughout the triplicate simulations, in contrast to the bare dihydropterin core. Their detailed behavior, however, was quite distinct because, in spite of similar initial docking positions, the two molecules ultimately assume different positions relative to NADPH: the guanidinium portion of molecule **32**, which bears a highly delocalized positive charge in a bulky substituent, sandwiches itself in the NADPH groove formed by the adenine heterocycle, the phosphates, and the nicotinamide, whereas the amino group in molecule **31**, with its concentrated positive charge on a small substituent, is instead attracted to the phosphate appended to the adenosine ribose C_3_, so that the ligand maneuvers itself towards the opposite face of the NADPH molecule (Figure 6). As a consequence, molecule **31** settles into a position where enough room is left between it and NADPH (Figure 7A) and folate can still approach the nicotinamide ring unimpeded (as shown in the upper panels of Figure 7), whereas a portion of molecule **32** always remains within 4 Å of the nicotinamide ring (Figure 6B), where it completely blocks the productive binding of folate (as shown in the lower panels of Figure 7).

To obtain estimates of the interaction strength between molecule **32** and R67 DHFR, umbrella sampling simulations of the separation process were performed. The interior of R67 DHFR forms a channel, and therefore two possible separation directions are possible: either through the “bottom” of the channel (where the adenine moiety of NADPH lies) or to the top of the channel (where the Lys32 residues involved in interactions of the benzoylglutamate chain in folate lie). Analysis of these results through the weighted-histogram analysis method (WHAM) showed that the complex is quite strongly bound, as ligand separation from its stable position through the bottom of the channel has an unfavorable barrier of approximately 12 kcal/mol (Figure 8).

### 2.5. Analysis of the Binding of Molecule ***32*** Able to Other Dihydrofolate Reductases

Chromosomally encoded prokaryotic dihydrofolate reductases have a very different overall structure and active site architecture from the plasmid-borne R67 DHFR: instead of a homotetrameric structure enclosing a full-length channel where NADPH and substrate bind, they are monomers where NADPH binds to a long surface cleft that leads into the active site pocket, to which the folate binds through an opening on the opposite side (Figure 9A). The differences in active site architecture are responsible for the different affinity of both enzymes to inhibitors such as trimethoprim, and it is therefore possible that a good inhibitor towards one of the forms will not be active towards the other. Docking our candidate molecules into chromosmal DHFR from *S. aureus* (3FRE) suggested that molecule **32** would be a better binder than the dihydrofolate core. Further analysis of the binding mode showed that the **32** binds in a peculiar way: whereas in dihydrofolate-bound protein, the long hydrophilic benzoylglutamate forces the pterin to orient its lactam-containing ring towards the bottom of the active site, in molecule **32** the bulky hydrophilic substituent (guanidinium) is attached to the lactam-containing ring substituent on the pterin ring, which forces the docking pose to have the opposite orientation of the pterin ring to ensure that the hydrophilic portion is oriented towards the open protein surface. This binding pose is, nonetheless, extremely stable: all triplicate simulations showed that the inhibitor remained very tightly bound to the active site, both through interactions with NADPH and with the active site residues (Figure 9). Attempts to separate the ligand from the active site by introducing an artificial force constant between the guanidinium and the bottom of the active site pocket and sequentially increasing the equilibrium distance of this artificial spring enabled us to compute the potential of mean force of the separation process (Figure 8). This value (11.5 kcal/mol) is very similar to the value obtained for ligand separation from the plasmid-borne DHFR, which suggests that this molecule can be used to control microorganisms bearing any of these types of dihydrofolate reductases.

The ability of candidate **32** to bind human DHFR was also analyzed (Table 2). Eukaryotic and prokaryotic chromosomal DHFR share a similar fold, and their structural differences are modest: an additional 10-aminoacid stretch between helix α1 and sheet β4, 11 aminoacids between α3 and β10 and an insertion of several aminoacids in the middle of the stretch what would otherwise form sheet β10 (Appendix A). The insertion between helix α1 and sheet β4 is responsible for the differences in the dynamic behavior observed upon substrate binding [21], and we observed that this subtle difference allows **32** to bind the eukaryotic enzyme in a manner similar to the one observed during folate binding (i.e., with the lactam ring of the pteridine pointing towards the bottom of the cavity). Triplicate molecular dynamics simulations showed that the binding of **32** is also very stable in this case, as it consistently remains in the pocket defined by the nicotinamide ring, the N-terminal of helix α1, and the β-sheets segments Ile-7-Val8 and Phe134-Tyr136 (Figure 10).

Comparison of the binding stability of **32** with that of the unmodified pteridine core **1** further highlighted the remarkable binding affinity of our novel ligand: the pteridine core **1** proved to be only weakly bound by the active site (Figure 11): in one of the simulations (Figure 11B) the ligand completely left the protein in less than 30 ns, whereas in each of the other two replicates the ligand assumed different binding modes: either moving (Figure 11C) towards Ile7 carbonyl (which keeps within a reactive distance relative to the nicotinamide ring), or in the opposite direction, towards Thr56, Ser59 and Val 115 (Figure 11D). The ability of folate to bind to the active site of human DHFR is therefore, (as in the R67 DHFR) not due to the pteridine core, but can be attributed instead to the favorable interactions of the *p*-aminobenzoylglutamate tail with the Arg70, Arg32 and Arg28 present at the entrance of the channel leading to the active site.

Finally, umbrella sampling simulations of the unbinding process of the complex between human-DHFR: **32** were performed to obtain the potential of mean force. The results (Figure 8) show that our novel ligand binds even more strongly to human DHFR than to the two bacterial DHFR, which raises the tantalizing possibility of its use in anti-cancer applications, like other human DHFR inhibitors like methotrexate.

## 3. Materials and Methods

### 3.1. Quantum Chemical Computations

Quantum chemical computation of the proton affinity and reduction energies of folate analogues were performed using with the Firefly quantum chemistry package [22], which is partially based on the GAMESS (US) source code [23]. The geometries of every analogue were optimized using the PBE0 density-theory functional [24,25] and 6-31+G(d) basis set with the help of autogenerated delocalized coordinates [26]. Single-point energies of the DFT-optimized geometries were then calculated using the same functional using the 6-311+G(2d,p). This particular combination of methods was earlier [27] shown to afford extremely small errors in protonation and reduction energies of NADH and other organic molecules. Solvation effects were computed using the Polarizable Continuum Model [28,29,30] implemented in Firefly.

### 3.2. Molecular Docking and Molecular Dynamics

Docking and molecular dynamics computations were performed in YASARA [31] using the 1.26 Å resolution crystal structure of R67 dihydrofolate reductase bound to NADPH and dihydrofolate (PDB: 2RK1) [20]. After excising the dihydrofolate ligand from the crystal structure, dihydrofolate-based ligands (Figure 1) were locally docked the wild-type structure with AutoDock 4.2.3 [32] using its Lamarckian genetic algorithm with default docking parameters and point charges assigned according to the AMBER14 force field [33]. The docking volume consisted of a 17.23 Å × 13.25 Å × 19.20 Å box centered on the crystallographic position of the dihydropterin moiety of the folate ligand. Selected ligands were also docked to the wild-type structure of human dihydrofolate reductase (PDB:4M6K) [21] bound to NADP^+^ and folate, after excising the folate ligand. The docking volume in this instance consisted of a 14.68 Å × 15.03 Å × 18.92 Å box centered on the crystallographic position of the dihydropterin moiety of the folate ligand. The docking box for *S. aureus* dihydrofolate reductase (PDB:3FRE) was a cube 77 Å wide that completely surrounded the whole protein. The ligands with higher dissociation energies were selected for further study through molecular dynamics. All molecular dynamics simulations were run with the AMBER14 forcefield [33], using a multiple time step of 1.25 fs for intramolecular and 2.5 fs for intermolecular forces. Simulations were performed in cubic cells at least 10 Å larger than the solute along each axis (65.9 Å wide for R67 DHFR, 72.00 Å wide for human DHFR), and counter-ions (22 Cl^−^ and 25 Na^+^ for R67 DHFR, 31 Cl^−^ and 33 Na^+^ for human DHFR) were added to a final concentration of 0.9% NaCl. In total, the simulation contained approximately 28,900 atoms (R67 DHFR) and 37,300 atoms (human DHFR). A 7.86 Å cutoff was taken for Lennard–Jones forces and the direct space portion of the electrostatic forces, which were calculated using the Particle Mesh Ewald method [34] with a grid spacing <1 Å, 4th order B-splines and a tolerance of 10^−4^ for the direct space sum. Simulated annealing minimizations started at 298 K, velocities were scaled down with 0.9 every ten steps for a total time of 5 ps. After annealing, simulations were run at 298 K. Temperature was adjusted using a Berendsen thermostat [35] based on the time-averaged temperature, i.e., to minimize the impact of temperature control, velocities were rescaled only about every 100 simulation steps, whenever the average of the last 100 measured temperatures converged. Substrate parameterization was performed with the AM1BCC protocol [36,37]. All simulations were run for at least 50 ns.

### 3.3. Computation of Potentail of Mean Force Using Umbrella Sampling/Wheighted Histogram Analysis Method

Umbrella sampling was performed on the complexes of molecule **32** with each DHFR to obtain an estimate of their relative binding energies. Fifteen windows were used for each system to sample the position of the ligand within the binding cavities. The centers of the umbrella potentials were spaced by 0.8 Å. In each window, a harmonic potential of the form V=1/2kx−x02 with a force constant of 5.0 kcal/mol/Å^2^ was used to apply the distance restraints between a ligand atom (generally chosen as the carbon atom in the guanidinium moiety of the ligand) and a protein atom (Cα Val31, for *S. aureus* DHFR; Cα Ile7, for human DHFR; Cα Gln67A, for R67 DHFR). For human DHFR, the ligand atom was instead chosen as the outermost methyl carbon of the methoxymethyl group on the ligand. Sampling was performed for 5.2 ns in each bin, of which the first 1.2 ns were used to enable the ligand to relax into the position defined by the new equilibrium distance of the harmonic potential and were therefore eliminated from the analyses. Statistics were thereafter collected every 0.5 ps. The potentials of mean force were obtained from the statistical distribution of the distances between the ligand and protein atoms in the restrained coordinate through the weighted histogram analysis method (WHAM) [38,39] using a bin size of 0.2 Å.

## 4. Conclusions

The docking and molecular dynamics studies above show that candidate **32** strongly binds to the to both types of prokaryotic dihydrofolate reductase, as well as to human DHFR. Its direct use as an antibiotic agent in clinical settings may therefore be expected to yield undesired side-effects through the inhibition of human cell metabolism. Such side-effects of its intended antibiotic use may be mitigated if **32** is instead used as a scaffold for the development of improved molecules that (for example) are unable to enter human cells or modifying it so that it would be quickly metabolized by liver enzymes upon intestinal absorption, so that it would never reach high concentrations in the systemic bloodstream. Further improvements of **32** in the direction of increased human toxicity (e.g., through enhanced binding caused by inclusion of the p-aminobenoylglutamate tail responsible for the large increase of affinity of dihydrofolate compared to the bare core **1**) may, on the other hand, afford better chemotherapeutic entities for use against fast-dividing, cancerous, cells. We expect this report to stimulate such developments towards the synthesis of novel anti-cancer drugs, as well as human-tolerated dual-acting antibiotics that target the two types of bacterial dihydrofolate reductase.

## Figures and Tables

**Figure 1 antibiotics-11-00779-f001:**
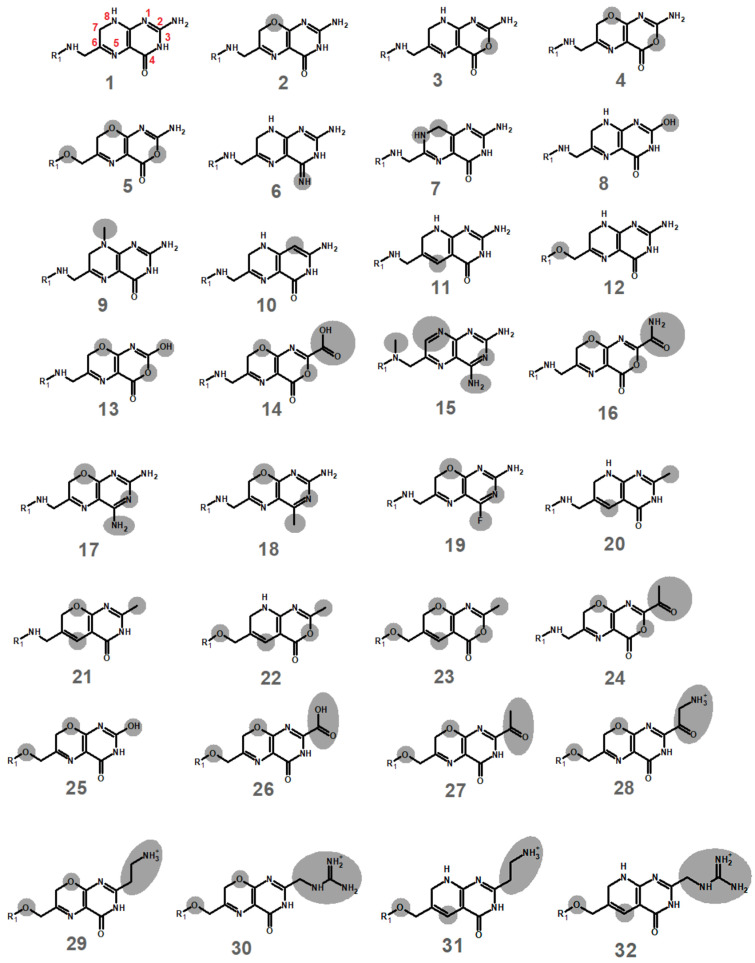
Putative inhibitors tested in this work. Differences from dihydrofolate (molecule **1**) are highlighted in grey. The pteridine core of methotrexate (which does not inhibit the plasmid-borne dihydrofolate reductase [3]) is depicted as molecule **15**.

**Figure 2 antibiotics-11-00779-f002:**
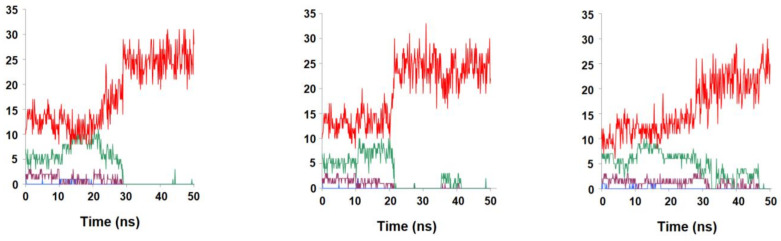
Evolution of contacts between the dihydropterin core (molecule **1**) and R67 DHFR along multiple simulations. red: number water molecules < 3 Å from the ligand; green: number of aminoacid residues < 3 Å from the ligand; violet: number of H-bonds between protein surface and the ligand; blue: number of H-bonds between NADPH and the ligand.

**Figure 3 antibiotics-11-00779-f003:**
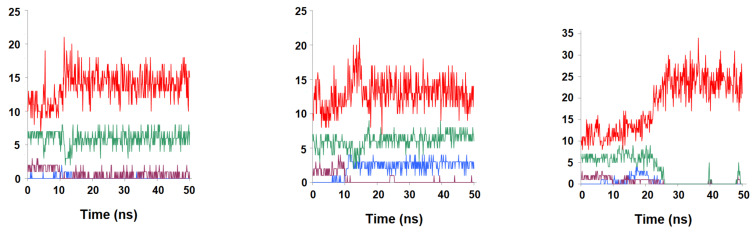
Evolution of contacts between the N^5^-protonated dihydropterin core (molecule **1**) and R67 DHFR along multiple simulations. red: number water molecules < 3 Å from the ligand; green: number of amino acid residues < 3 Å from the ligand; violet: number of H-bonds between protein surface and the ligand; blue: number of H-bonds between NADPH and the ligand.

**Figure 4 antibiotics-11-00779-f004:**
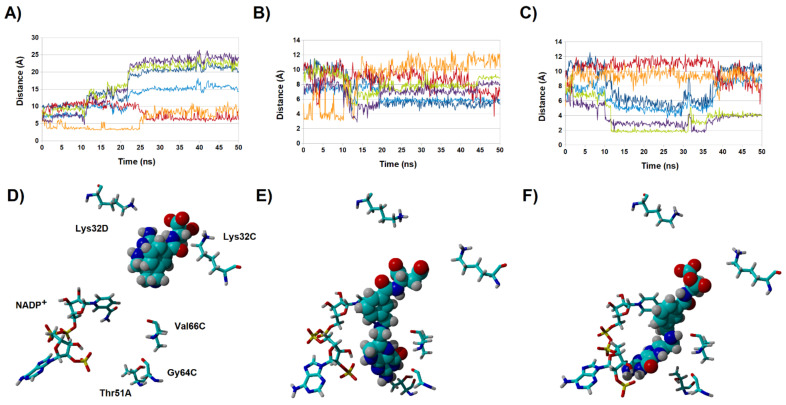
Evolution of key distances between N^5^-deprotonated dihydrofolate and R67 DHFR along multiple simulations (**A**–**C**). Dark orange: distance between 2′-phosphate and folate N^3^-H; violet: distance between Gly64 C=O and folate N^3^-H; light blue: distance between Val66 C=O and folate N^8^-H; dark blue: distance between nicotinamide C^4^ and folate C^6^; green: distance between Val66 NH and folate C=O; violet: distance between Thr51A C=O and folate NH_2_ group. (**D**–**F**): the final poses obtained at the end of each 50 ns simulation. Most amino acids have been hidden for clarity.

**Figure 5 antibiotics-11-00779-f005:**
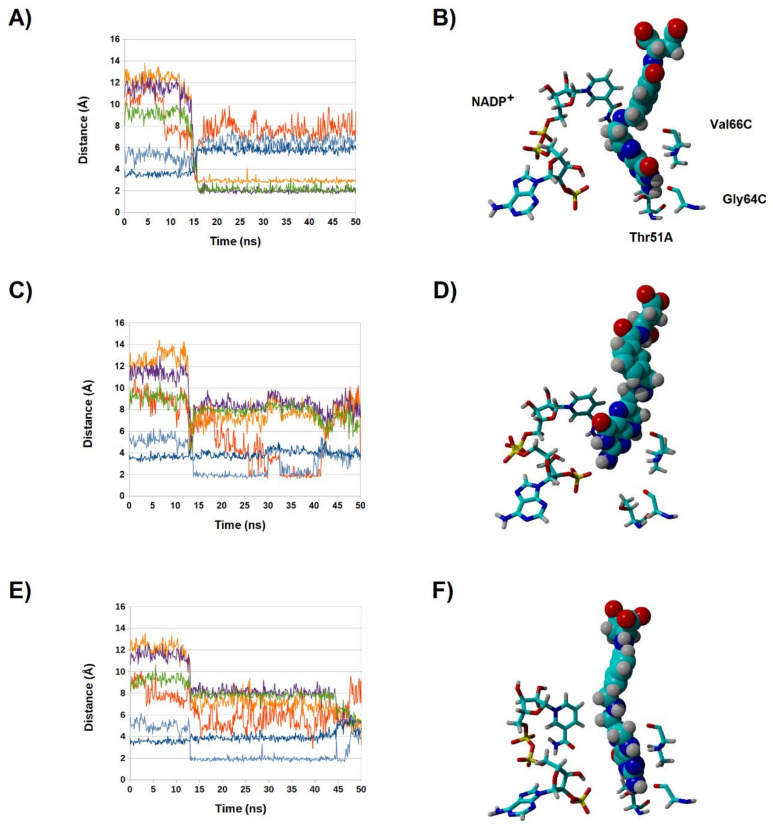
Evolution of key distance between N^5^-protonated dihydrofolate and R67 DHFR along multiple simulations (**A**,**C**,**E**). Light-green: distance between 2′-phosphate and folate N^3^-H; violet: distance between 2′-phosphate and folate NH_2_ group; light blue: distance between Val66 NH and folate N^1^; dark blue: distance between Gly64 C=O and folate NH_2_; orange: distance between Lys32C and folate glutamyl carboxylate; red: Lys32D and folate glutamyl carboxylate. (**B**,**D**,**F**): the final poses obtained at the end of each 50 ns simulation. Most amino acids have been hidden for clarity.

**Figure 6 antibiotics-11-00779-f006:**
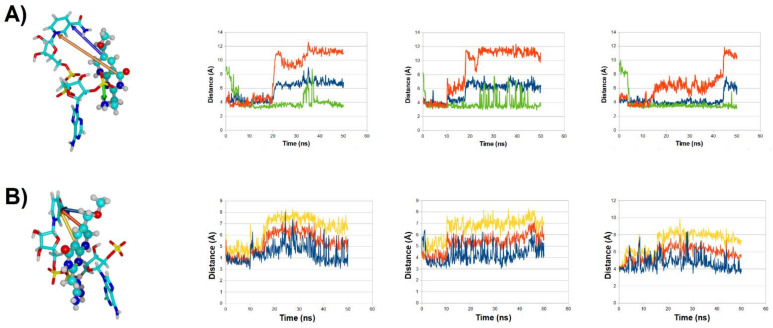
Binding modes and evolution of key ligand-NADPH distances along three molecular dynamics simulations for ligand **31** (**A**) and ligand **32** (**B**). NADPH is depicted as sticks, ligands as solid atom sphere models. Each line in the graph corresponds to the evolution of the distance between the atoms connected with an arrow of the same color in the left-most panel.

**Figure 7 antibiotics-11-00779-f007:**
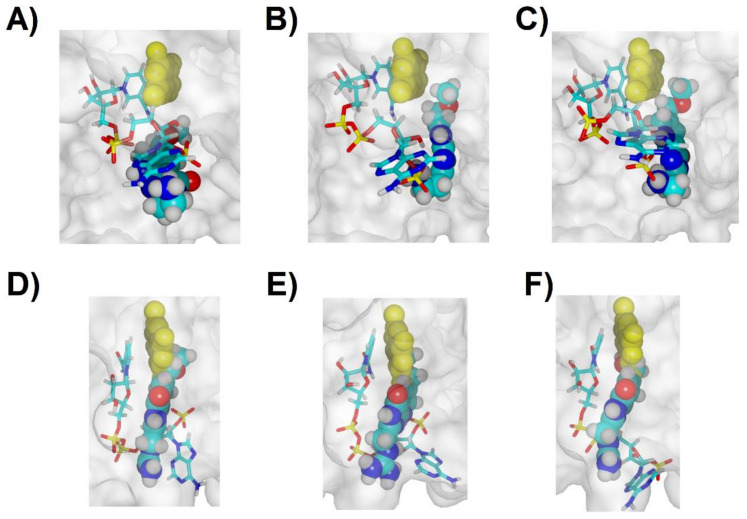
Final binding modes (after 50 ns) of molecule **31** (**A**–**C**): three replicate simulations, shown in the same relative orientation; and molecule **32** (**D**–**F**): three replicate simulations, shown in the same relative orientation, rotated ≈ 90° relative to the orientation of panels (**A**–**C**). NADPH is depicted as sticks, ligands as solid atom sphere models. In each panel, the crystallographic position (PDB:2RK1) of the dihydropterin core of folate is depicted as yellow solid atom sphere models.

**Figure 8 antibiotics-11-00779-f008:**
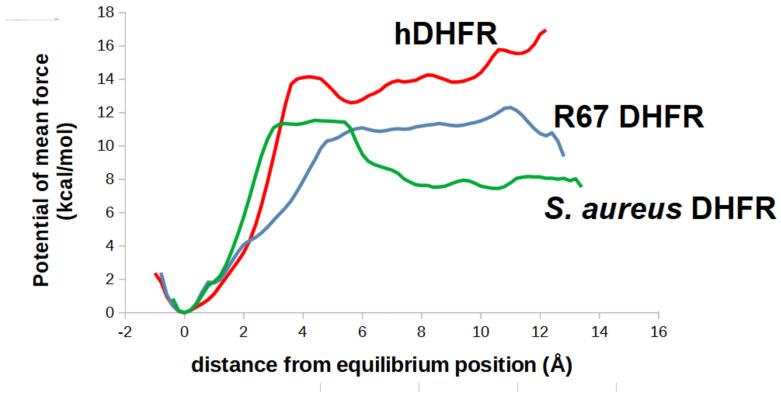
Potentials of mean force of the separation of ligand **32** from each of the three examined dihydrofolate reductases.

**Figure 9 antibiotics-11-00779-f009:**
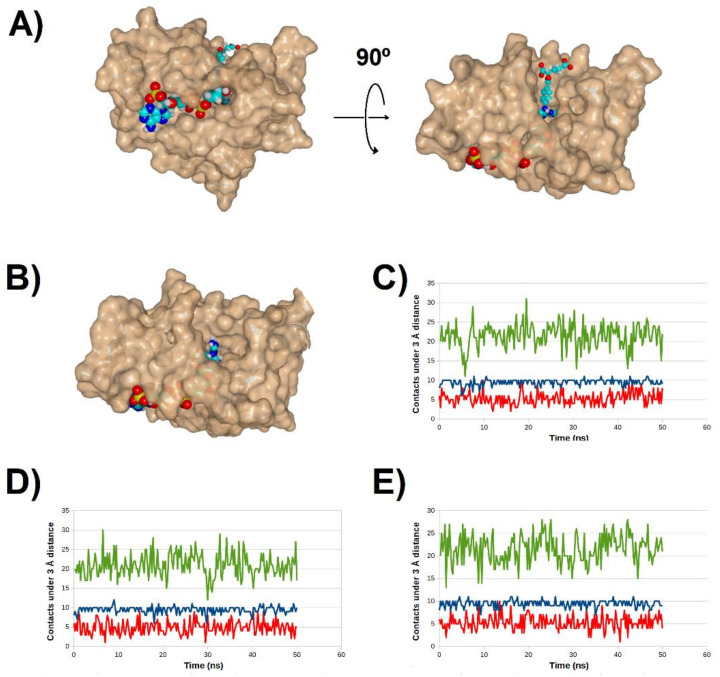
Comparing binding modes of chromosomally encoded prokaryotic dihydrofolate reductase towards dihydrofolate and molecule **32**. (**A**) Active site architecture of dihydrofolate-bound DHFR from *S. aureus* (PDB:3FRD); (**B**) final binding mode (after 50 ns) of molecule **32**. Protein orientation is the same as in the right-most panel of Figure 8A; (**C**–**E**): evolution of number of contacts between molecule **32** and NADPH/protein in three replicate simulations: green: number of protein atoms within 3 Å of ligand; red: number of NADPH atoms within 3 Å of ligand; blue: number of residues within 3 Å of ligand.

**Figure 10 antibiotics-11-00779-f010:**
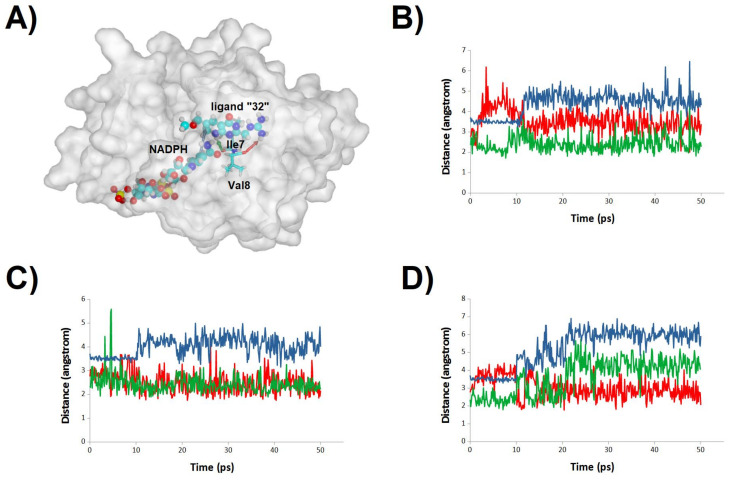
Stability of the binding mode of molecule **32** at the human DHFR active site. (**A**) Representative snapshot of a trajectory of ligand-bound human dihydrofolate-bound DHFR; (**B**–**D**) evolution of key ligand-protein distances in three simulations. Each line in the graph corresponds to the evolution of the distance between the atoms connected with an arrow of the same color in panel (**A**): blue for NADPH:ligand distance, red for the Val8 carbonyl:guanidinium distance and green for the Ile7 carbonyl:ligand N^1^-atom distance.

**Figure 11 antibiotics-11-00779-f011:**
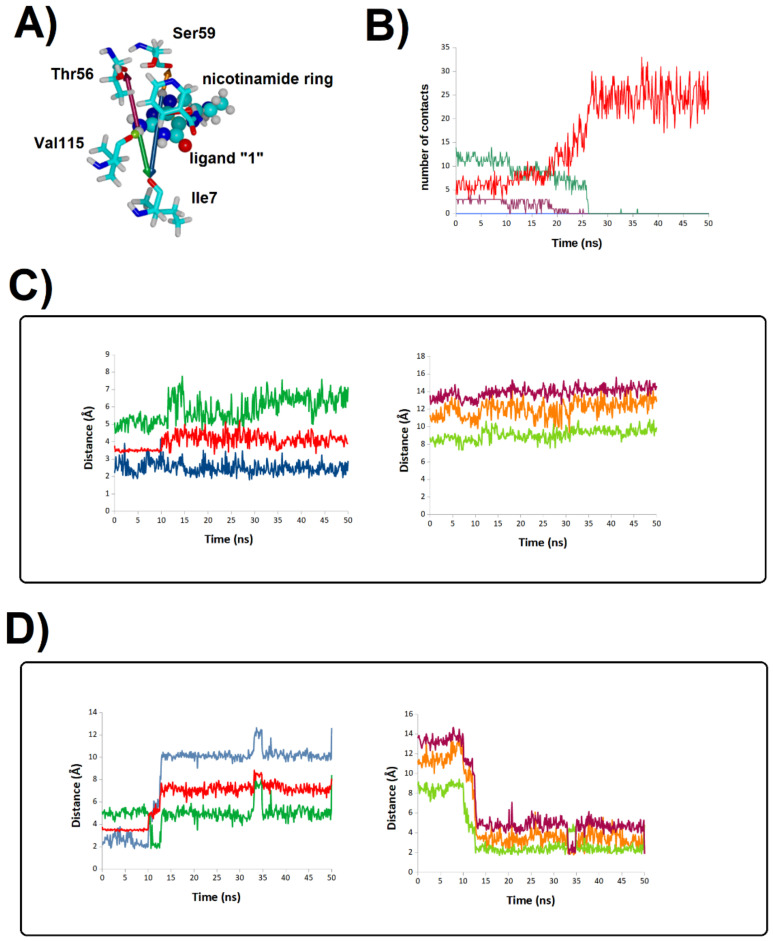
Stability of the binding mode of molecule **1** at the human DHFR active site. (**A**) Representative snapshot of a trajectory of pteridince core-bound human dihydrofolate-bound DHFR; (**B**) evolution of ligand-protein contacts in the simulation where unbinding was observed. Red: number water molecules < 3 Å from the ligand; green: number of amino acid residues < 3 Å from the ligand; violet: number of H-bonds between protein surface and the ligand; blue: number of H-bonds between NADPH and the ligand. (**C**,**D**) Evolution of key ligand-protein distances in the remaining two simulations. Each line in the graph corresponds to the evolution of the distance between the atoms connected with an arrow of the same color in panel (**A**).

**Table 1 antibiotics-11-00779-t001:** PBE0/6-311+G(2d,p)//PBE0/6-31+G(d) energies for the protonation (at position 5) and hydride transfer from NADH to each molecule (in its original protonation state). Bolded values show molecules significantly less basic than the pteridine core (>10 kcal·mol^−1^ difference) or with reduction energies above 22 kcal·mol^−1^.

Molecules	Protonation Energy (kcal/mol) (vs. Dihydrofolate Core)	Reduction by NADH (kcal/mol)	Molecule	Protonation Energy (kcal/mol) (vs. Dihydrofolate Core)	Reduction by NADH (kcal/mol)
**1**	0.0	**35.2**	**17**	**10.6**	**28.0**
**2**	4.0	**28.8**	**18**	8.2	**26.2**
**3**	2.8	**33.5**	**19**	**10.5**	**22.9**
**4**	7.0	**26.3**	**20**	**20.1**	**46.0**
**5**	9.4	22.0	**21**	**26.0**	**37.2**
**6**	−2.9	**38.4**	**22**	**25.1**	**40.2**
**7**	−9.1	**39.6**	**23**	**33.8**	**32.2**
**8**	1.8	**33.2**	**24**	7.9	4.5
**9**	−0.4	**36.3**	**25**	8.2	20.3
**10**	−2.4	**37.8**	**26**	**11.4**	2.3
**11**	**13.9**	**53.8**	**27**	**11.7**	3.6
**12**	2.2	**30.5**	**28**	**16.1**	2.2
**13**	9.8	**23.8**	**29**	8.6	16.1
**14**	9.0	5.8	**30**	**11.7**	16.2
**15**	**16.8**	**40.7**	**31**	**20.7**	**42.8**
**16**	8.7	10.4	**32**	**24.1**	**39.7**

**Table 2 antibiotics-11-00779-t002:** Dissociation energies (kcal·mol^−1^) of selected molecules from the active sites of selected dihydrofolate reductases, computed using the Autodock scoring functions on the highest-scoring docking poses.

Molecules	R67 DHFR (PDB: 2RK1)	Human DHFR (PDB: 4M6K)	*S. aureus* DHFR (PDB: 3FRE)
**1**	5.85	6.58	5.57
**11**	5.81	6.61	5.03
**15**	5.00	4.39	4.55
**20**	6.29	6.93	5.81
**21**	6.76	7.42	5.97
**22**	5.91	6.40	5.76
**23**	6.01	6.76	6.09
**31**	7.94	8.40	7.89
**32**	6.94	7.78	6.70

## Data Availability

The outputs of the quantum chemical computations and snapshots of all MD simulations have been deposited in Figshare https://doi.org/10.6084/m9.figshare.19782616 (accessed on 17 May 2022).

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
