# Peer review of "Computational Development of Inhibitors of Plasmid-Borne Bacterial Dihydrofolate Reductase"

_antibiotics, 2022, doi:10.3390/antibiotics11060779_

Round 1

Reviewer 1 Report

Silva describes a computational project designing inhibitors. DFT calculations are verified using MD simulations. The report is well-written, and the data support the conclusions.

I am, in general, supportive of the publication of this manuscript; however, I would like to see a much more extensive description of the methods and the data provided via figshare:

  • The data provided via figshare is a good step in the right direction. However, I am missing a README file with some information about the provided data.
  • While the input files for firefly are provided; the input files for the other programs are missing; please provide these where possible
  • re-structure the method section with separate sections for each method
  • I am missing descriptions for several methods, for example:
    • On line 281, Silva writes: "introducing an artificial force constant 281 between the guanidinium and the bottom of the active site pocket and sequentially 282 increasing the equilibrium distance of this artificial spring enabled us to compute the 283 potential of mean force of the separation process." - please describe this and how the result, 11.5 kcal/mol, was calculated.
    • Many figures contain data extracted from the MD trajectories - please describe how this data was extracted.

Author Response

Silva describes a computational project designing inhibitors. DFT calculations are verified using MD simulations. The report is well-written, and the data support the conclusions.

I am, in general, supportive of the publication of this manuscript; however, I would like to see a much more extensive description of the methods and the data provided via figshare:

The data provided via figshare is a good step in the right direction. However, I am missing a README file with some information about the provided data.

I have now included a full description of the contents of the repository in the respective homepage.

While the input files for firefly are provided; the input files for the other programs are missing; please provide these where possible

I have now, as requested, uploaded the input files and scripts for the molecular dynamics simulations and the analysis of trajectories.

re-structure the method section with separate sections for each method

Done, as requested.

I am missing descriptions for several methods, for example:

On line 281, Silva writes: "introducing an artificial force constant 281 between the guanidinium and the bottom of the active site pocket and sequentially 282 increasing the equilibrium distance of this artificial spring enabled us to compute the 283 potential of mean force of the separation process." - please describe this and how the result, 11.5 kcal/mol, was calculated.

In lines 420-436 I have now included more detail on the umbrella-sampling/WHAM method used for these computations.

Many figures contain data extracted from the MD trajectories - please describe how this data was extracted.

These data were extracted using the analysis macros that are now deposited in figshare. In short, the “default” analysis script included in the YASARA package was selectively modified to measure, for each saved point in the trajectory, the values of key ligand-protein distances, number of aminoacids (or water molecules) within a given distance of the ligand, etc.

Reviewer 2 Report

Pedro J. Silva in this manuscript described the process of designing R67 DHFR-targeting drug candidates. The author showed the best candidate “molecule 32” can bind both bacteria and human DHFR strongly and this drug has the potential for chemotherapeutic agents. While this work could be refined to better show the story.

1. From lines 59-62, the author stated that because of the structural difference between R67 DHFR and classical DHFR, the R67-DHFR-targeting drug should have low toxicity to human cells. However, the final candidate molecule 32 even has a stronger binding ability to human DHFR. The author should consider re-editing sentences here to make the logic more coherent.

2. The DHFR-targeting drug has the potential to be used for antibiotics and human chemotherapeutic agents. The author should describe the difference in selection criteria of drugs to be used as antibiotics or chemotherapeutic agents.

3 From lines 123-126, the author mentioned that molecules 31 and 32 were the most likely to afford strong binding than the dihydropterin core of dihydrofolate for all three types of DHFR. Since the initial purpose is to identify the R67-DHFR-targeting drug, why did the author not select molecules that show strong binding ability toR67-DHFR and low binding ability to human-DHFR?

4. The author focused on three types of DHFR: R67, human, and S.aureus DHFRs. It would be clearer that sequences of amino acids of three types of DHFR are aligned and annotated with their structures in the sequences. Then, the relationship between amino acids sequence conservation and drug potential targeting sites will be clear.

5. Except for plasmid-encoded R67-DHFR, bacteria have other mechanisms for trimethoprim-resistance, such as F98Y DHFR mutant from S. aureus with RAB1. The author may introduce some more trimethoprim-resistance mechanisms in the introduction.

Bourne, Christina R., et al. "Inhibition of antibiotic-resistant Staphylococcus aureus by the broad-spectrum dihydrofolate reductase inhibitor RAB1." Antimicrobial agents and chemotherapy 54.9 (2010): 3825-3833.

6. The abbreviation DHRS is not used for many dihydrofolate reductases in the manuscript.

7. The author mentioned Density-functional calculations in the abstract. It was listed as a keyword as well. Not clear where the density-functional calculation is used in the analysis.

Author Response

(reviewer's text in black, my responses in red)

Pedro J. Silva in this manuscript described the process of designing R67 DHFR-targeting drug candidates. The author showed the best candidate “molecule 32” can bind both bacteria and human DHFR strongly and this drug has the potential for chemotherapeutic agents. While this work could be refined to better show the story.

1. From lines 59-62, the author stated that because of the structural difference between R67 DHFR and classical DHFR, the R67-DHFR-targeting drug should have low toxicity to human cells. However, the final candidate molecule 32 even has a stronger binding ability to human DHFR. The author should consider re-editing sentences here to make the logic more coherent.

Lines 59-62 state that such molecules are “likely” to afford lower toxicity. This, however, is not guaranteed. I chose the expression “is likely to afford” to hint at that, but I suppose that this was not as clear as I thought. I have now rephrased this expression to “may enable”. I hope this change makes the strategy (and its limitations) more apparent.

2. The DHFR-targeting drug has the potential to be used for antibiotics and human chemotherapeutic agents. The author should describe the difference in selection criteria of drugs to be used as antibiotics or chemotherapeutic agents.

Such criteria are hard to generalize beyond the “obvious” ones (i.e. anticancer agents should be able to enter eukaryotic cells and bind to enzymes that are especially active in fast-dividing cells; antibacterials should be able to enter prokaryotic cells and either destabilize their cell walls or inhibit key enzymes in prokaryotic metabolism). Developing a “dual candidate” into chemotherapeutic use might involve adapting it to prevent its uptake by prokaryotic cells, and (conversely) decreasing its cytotoxic effects towards human cells might be achieved by preventing it from being transported into eukaryotic cells, or modifying it so that it would be quickly metabolized by liver enzymes upon intestinal absorption, so that it would never reach high concentrations in the systemic bloodstream. I am afraid that it is hard to be more precise without runnig the risk of falling prey to flawed generalizations. I have now added this information to the “Conclusions” section.

3 From lines 123-126, the author mentioned that molecules 31 and 32 were the most likely to afford strong binding than the dihydropterin core of dihydrofolate for all three types of DHFR. Since the initial purpose is to identify the R67-DHFR-targeting drug, why did the author not select molecules that show strong binding ability toR67-DHFR and low binding ability to human-DHFR?

My initial aim was to find good candidates for binding to the bacterial (especially R67) DHFR. Analysis of the ligands’ ability to target human DHFR was a later addition to my research, and aimed at ascertaining if selectivity had indeed been achieved and to make sure I was not unwittingly “over-selling” the potential of my molecules. I was surprised to find that the correlation between the affinity of the ligands for R67 DHFR and human DHFR was quite high (R=0.934), and therefore there were no candidates with low affinity towards human DHFR and high affinity towards R67 DHFR. Any attempt to analyse a candidate with lower affinity to the human enzyme would necessarily entail a corresponding high likelihood of ending up with a candidate with limited ability to bind the target I was originally interested in (R67 DHFR).

4. The author focused on three types of DHFR: R67, human, and S.aureus DHFRs. It would be clearer that sequences of amino acids of three types of DHFR are aligned and annotated with their structures in the sequences. Then, the relationship between amino acids sequence conservation and drug potential targeting sites will be clear.

R67 DHFR has no sequence or structural similarity with the other DHFR. I have now provided (as Supporting information) a figure comparing the structures and sequences of human and S. aureus DHFR.

5. Except for plasmid-encoded R67-DHFR, bacteria have other mechanisms for trimethoprim-resistance, such as F98Y DHFR mutant from S. aureus with RAB1. The author may introduce some more trimethoprim-resistance mechanisms in the introduction. (Bourne, Christina R., et al. "Inhibition of antibiotic-resistant Staphylococcus aureus by the broad-spectrum dihydrofolate reductase inhibitor RAB1." Antimicrobial agents and chemotherapy 54.9 (2010): 3825-3833.)

The reviewer is right that resistance mechanisms other than R67 exist. Those have been hinted at in line 36, and I am afraid that discussing them in more detail might detract from the focus of the paper, since I have not analyzed the binding of ligand 32 to those other mutants.

6. The abbreviation DHRS is not used for many dihydrofolate reductases in the manuscript.

I am afraid I do not quite understand what the reviewer means here: I have not used DHRS as an abbreviation.

7. The author mentioned Density-functional calculations in the abstract. It was listed as a keyword as well. Not clear where the density-functional calculation is used in the analysis.

The quantum chemical method used was a density-functional theory method. I have now added that information in the Methods Section.

Round 2

Reviewer 1 Report

The author present a revised manuscript in which all my concerns have been addressed. I, hence, recommend Antibiotics to consider this manuscript for publication.